# Shifts in *Fusarium* Communities and Mycotoxins in Maize Residues, Soils, and Wheat Grains throughout the Wheat Cycle: Implications for Fusarium Head Blight Epidemiology

**DOI:** 10.3390/microorganisms12091783

**Published:** 2024-08-28

**Authors:** Toan Bao Hung Nguyen, Amandine Henri-Sanvoisin, Monika Coton, Gaétan Le Floch, Adeline Picot

**Affiliations:** Univ Brest, INRAE, Laboratoire Universitaire de Biodiversité et Écologie Microbienne, F-29280 Plouzané, France; toan-bao-hung.nguyen@univ-brest.fr (T.B.H.N.); amandine.henri@univ-brest.fr (A.H.-S.); monika.coton@univ-brest.fr (M.C.); gaetan.lefloch@univ-brest.fr (G.L.F.)

**Keywords:** metabarcoding, qPCR, *Fusarium graminearum*, *F. poae*, *F. avenaceum*, diversity, dynamics

## Abstract

Fusarium Head Blight (FHB), predominantly caused by *Fusarium* species, is a devastating cereal disease worldwide. While considerable research has focused on *Fusarium* communities in grains, less attention has been given to residues and soil, the primary inoculum sources. Knowledge of *Fusarium* spp. diversity, dynamics, and mycotoxin accumulation in these substrates is crucial for assessing their contribution to wheat head infection and the complex interactions among *Fusarium* communities throughout the wheat cycle. We monitored six minimum-tillage wheat fields, with maize as the preceding crop, over two years. Soils, maize residues, and wheat grains were sampled at four stages. *Fusarium* composition was analyzed using a culture-dependent method, species-specific qPCR, and *EF1α* region metabarcoding sequencing, enabling species-level resolution. The *Fusarium* communities were primarily influenced by substrate type, accounting for 35.8% of variance, followed by sampling location (8.1%) and sampling stage (3.2%). Among the 32 identified species, *F. poae* and *F. graminearum* dominated grains, with mean relative abundances of 47% and 29%, respectively. Conversely, residues were mainly contaminated by *F. graminearum*, with a low presence of *F. poae*, as confirmed by species-specific qPCR. Notably, during periods of high FHB pressure, such as in 2021, *F. graminearum* was the dominant species in grains. However, in the following year, *F. poae* outcompeted *F. graminearum*, resulting in reduced disease pressure, consistent with the lower pathogenicity of *F. poae*. Source Tracker analysis indicated that residues were a more significant source of *Fusarium* contamination on wheat in 2021 compared to 2022, suggesting that *F. graminearum* in 2021 primarily originated from residues, whereas *F. poae*’s sources of infection need further investigation. Additionally, multiple mycotoxins were detected and quantified in maize residues during the wheat cycle, raising the question of their ecological role and impact on the soil microbiota.

## 1. Introduction

Fusarium Head Blight (FHB), caused by different *Fusarium* and *Microdochium* species, is one of the most devastating fungal diseases in cereals worldwide [1]. Beyond reducing grain yield and quality, FHB poses serious safety concerns due to the *Fusarium* species’ ability to produce mycotoxins that persist throughout the food chain [2]. These secondary or specialized metabolites can be toxic to humans and animals, causing adverse effects on the liver, kidney, immune system, and nervous system [3,4], prompting regulatory thresholds for cereal grains and cereal-based products (e.g., regulation EU 2023/915 in Europe). At least sixteen *Fusarium* species, predominantly *F. graminearum*, *F. culmorum*, *F. avenaceum*, and *F. poae*, were reported to be responsible for the FHB of small-grain cereals [4,5]. According to species, aggressiveness, mycotoxin production profiles, geographic distribution, host range, and climate preference can greatly vary [6,7]. *Fusarium* species often co-occur in plants [8,9], resulting in cereal products simultaneously contaminated by multiple mycotoxins, ranging from the most dominant and regulated mycotoxins (fumonisins, trichothecenes, and zearalenone) [4] to emerging mycotoxins (beauvericin, enniatins, moniliformin, and fusaproliferin) [10,11]. These multi-mycotoxin contamination patterns have raised public health concerns over the last decade due to potential synergistic or additive toxicities impacting human or animal health [12,13].

Co-occurrence patterns imply that both disease development and mycotoxin production are influenced by interactions among members of the *Fusarium* communities as well as with the surrounding environment, including the plant host and phytomicrobiome [7,14,15,16]. In this regard, FHB well illustrates the pathobiome concept, defined as “the set of host-associated organisms associated with reduced (or potentially reduced) health status as a result of interactions between members of that set and the host” [17]. Therefore, monitoring *Fusarium* at the community level, rather than focusing on one species, is essential for efficient and tailored disease management strategies. Characterizing the FHB species complex often relies on culture and molecular-based methods, such as fungal isolations on selective media, multiplex, or real-time qPCR for species identification and/or quantification. More recently, metabarcoding-based approaches have also been developed, mostly targeting the elongation factor (*EF1α*) for species-level resolution to describe *Fusarium* communities in environmental samples such as maize residues and soil [18].

*Fusarium* spp. infections in wheat heads primarily occur during flowering and are influenced by factors such as inoculum load in the field, weather conditions, wheat cultivar susceptibility, and agronomic practices [5,19,20]. Main inoculum sources include airborne spores or splash-dispersed conidia originating from previously infected host residues, such as maize and small-grain cereals [21,22,23]. *Fusarium* spp. can survive saprophytically on previous crop residues, discharging highly resistant ascospores and conidia both locally and on a more distant scale [24,25]. The amount of residue left from preceding crops, in particular maize stalks and kernels, significantly increases FHB severity and deoxynivalenol (DON) contamination in future crops [26,27]. Recent studies indicated that maize as a preceding crop, combined with no-tillage or minimum tillage, leads to high disease levels in wheat grains [28,29]. Past FHB management recommendations included burying *Fusarium*-infected residues and crop rotations with non-host species [30]. Although tillage significantly reduces FBH incidence, soil conservation practices, including reduced tillage, have substantially increased in the past few decades in various countries in an effort to prevent soil erosion and degradation [31,32].

Despite the fact that crop residues and soil have long been considered the main reservoirs of *Fusarium* spp., accurate descriptions of *Fusarium* community dynamics in these substrates are scarce. Such data are crucial for accurately determining the predominant *Fusarium* spp. in agroecosystems and their contribution to wheat head infection with FHB. This knowledge will also help better understand the complex interactions between *Fusarium* spp. that are at stake during disease onset and development, as well as conditions leading to mycotoxin production. Furthermore, *Fusarium*-infected residues are major diffusive sources of mycotoxins in the environment, including soil and drainage water [33]. The diversity, role, and fate of mycotoxins in maize residues left on the soil surface throughout the wheat cycle under natural conditions are scarcely investigated. The objective of this study was to thoroughly investigate *Fusarium* community diversity, dynamics, and mycotoxin contamination in soils, maize residues, and wheat grains. We monitored six agricultural wheat fields at four time points throughout the wheat cycle over two years. We used a combined approach including *EF1α* region metabarcoding sequencing, culture-dependent analyses, species-specific quantitative PCR, and *Fusarium*-related mycotoxin quantifications using HR-QTOF-LC/MS. The 15 targeted mycotoxins included deoxynivalenol (DON), 15-acetyl-deoxynivalenol (15ADON), 3-acetyl-deoxynivalenol (3ADON), zearalenone (ZEA), nivalenol (NIV), moniliformin (MON), beauvericin (BEA), T-2, HT-2, enniatin A (ENNA), A1 (ENNA1), B (ENNB), B1 (ENNB1), fumonisin B1 (FB1), and B2 (FB2). Relationships between FHB disease, mycotoxin contamination levels, *Fusarium* community composition, and climatic conditions were also examined to enhance understanding of FHB epidemiology and interactions among causal agents.

## 2. Materials and Methods

### 2.1. Field Sampling

Six soft winter wheat fields (F1 to F6) from six different farms in Brittany, France (Figure 1), were selected for a 2-year FHB monitoring survey (2020–2021, designated as Y1 and 2021–2022, designated as Y2) (Table 1). Due to crop rotation, the fields studied were different from year to year but remained under the same farmer’s management, except for F5Y2, which was located approximately 10 km from F5Y1. Weather data for each field, including cumulative monthly precipitation, mean daily humidity, and temperature during the grain filling (including one week before and two weeks after the estimated date of flowering) and maturity period (from one week before flowering to harvest), were collected from the nearest weather station (MétéoFrance data). All fields were cultivated using practices that maximized the FHB risk, specifically maize-wheat rotation (either maize silage or maize grain as a previous crop) and minimum tillage. This latter practice was the standard procedure before wheat sowing, except under exceptional circumstances requiring farmers to plough their field (F1 in 2020 and F2 in 2021). All fields were planted with either a single or a mixture of soft winter wheat varieties. Three sampling points with GPS coordinates were defined per field, ensuring that samples were collected from the exact same zones at each time point. For each sampling point, a composite sampling method was used, involving 10 sub-samples per sampling point within a 5 m radius half-circle. The first 5 cm of soil and the aboveground maize residue fractions were collected after maize harvest (in October, T1), at seedling (in January, T2), 2 to 3 weeks after flowering (in June, T3, with flowering defined as the beginning of anther extrusion), and at harvest (in July, T4). Wheat heads were collected 2 to 3 weeks after flowering, and grains were collected at harvest. Soil samples, after removing stones or plant residues, were stored at −80 °C before DNA extraction. Residue and wheat head samples were stored at −80 °C, then lyophilized and ground using a Retsch MM400 Mixer Mill (Retsch, Haam, Germany) before DNA extraction. All samples were processed within 1 to 3 days after field sampling.

### 2.2. Disease Evaluation at Flowering and Harvest

FHB symptoms were evaluated 2 to 3 weeks after flowering in 100 wheat heads in each of the three sampling points per field on a scale of 0 to 4, where 1 = <10, 2 = <25, 3 = <75, and 4 = 100% of spikelets exhibited presumptive FHB symptoms, typically premature bleaching with a pinkish to orangish coloration (Figure 2). The FHB severity index (%) was calculated as the product of incidence and severity [35], where incidence is the percentage of infected heads (number of infected heads/total number of heads scored) and severity is the mean score of infected heads (∑ (number of heads × score in %)/number of infected heads).

The percentage of grains contaminated by *Fusarium* at harvest was evaluated in 10 and 13 wheat heads per sampling point in 2021 and 2022, respectively, by a culture-dependent method. Five wheat grains per head were placed on potato dextrose agar (PDA) supplemented with 50 mg L^−1^ of streptomycin, penicillin, and chlortetracycline, then incubated at room temperature under natural sunlight exposure for 5–7 days. The percentage of grains contaminated by *F. graminearum*, *F. poae*, *F. avenaceum*, and *Microdochium* spp. was assessed by morphological identification and by spore microscopic observations when needed.

### 2.3. DNA Extraction

Total DNA from soil samples was extracted using 2 × 0.5 g of soil with the NucleoSpin^®^ Kit for Soil (Macherey-Nagel, Duren, Germany) according to the manufacturer’s instructions. For maize residues, wheat heads, and grains, DNA was extracted from 100 mg of dried ground samples using the FastDNA™ SPIN kit (MP Medicals, Solon, OH, USA), following the manufacturer’s instructions. DNA quality and concentration were determined using a UV Spectrophotometer (Nanodrop1000, Thermo Fisher Scientific, Waltham, MA, USA). DNA was stored at −20 °C until use.

### 2.4. Illumina MiSeq Sequencing

Primers Fa-150 (5′-CCGGTCACTTGATCTACCAG-3′) and Ra-2 (5′-ATGACGGTGACATAGTAGCG-3′) were used to amplify *EF1α* regions of *Fusarium* species [18]. Amplicon libraries and Illumina MiSeq PE300 sequencing were performed at the Génome Québec Innovation Centre, Montreal, Canada. Adapter FLD_ill (forward sequence: 5‘-ACACTCTTTCCCTACACGACGCTCTTCCGATCT-3′ and reverse sequence: 5′-GTGACTGGAGTTCAGACGTGTGCTCTTCCGATCT-3′) was used. PCR mixtures in an 8 µL total volume included 1X Qiagen Buffer, 1.5 mM MgCl_2_, 5% DMSO, 0.2 mM dNTP mix, 0.01 U µL^−1^ Qiagen HotStarTaq, 0.6 µM of each primer, and 8 pM DNA. PCR conditions were 15 min at 96 °C, followed by 35 cycles of 30 s at 96 °C, 30 s at 52 °C, and 60 s at 72 °C, and a final extension of 10 min at 72 °C. The sequencing of samples from 2021 (Y1) was split into two separate Illumina runs, with 84 samples in run 1 and 87 in run 2. In 2022 (Y2), all samples were combined and sequenced in two separate runs. In each case, the sequencing depth was 25,000 reads per sample.

### 2.5. Processing of Sequencing Data

The processing of sequencing data was performed by combining data from Y1 with that from Y2. Raw sequencing data were processed using the pipeline developed by Cobo-Díaz et al. (2019), with a few modifications [18]. Briefly, the quality profiles of reads were first inspected using plotQualityProfile function from DADA2 pipeline [36], using default parameters except for truncation length. These trimming parameters were determined beforehand (forward and reverse trim positions were 269 and 221 bp, respectively; expected error values were 3 for both) by Figaro version 1.16 [37], with the estimated amplicon length set to 430 bp and the minimum overlap length set at 20 bases. Then, forward and reverse read pairs were trimmed, denoised, and filtered using the DADA2 pipeline [36]. Amplicon sequence variants (ASV) were independently inferred from the forward and reverse using the run-specific error rates, and read pairs were merged with at least 12 bp overlap. Chimeras were also removed using the removeBimeraDenovo function of DADA2. The average percentages of read retention were 81.2% after filtering, 80.9% after denoising, 75.4% after merging, and 68.7% after chimera removal. As the sequencing of samples from Y2 was split into two runs, ASV counts were summed for these samples. Taxonomic assignment was performed using BlastN against the NCBI Genbank nr/nt sequence database. Non-Nectriaceae sequences with less than 95% identity and/or less than 380 bp length were removed, representing 25.3% of total ASVs. Unassigned taxa and ASVs whose minimum count across the dataset was less than 10 were also removed using filter_taxa_from_otu_table.py and filter_otus_from_otu_table.py functions of QIIME [38]. Relative abundances were calculated at the ASVs and species level.

### 2.6. qPCR Quantification of Dominant Fusarium Species

*Fusarium graminearum*, *F. poae*, and *F. avenaceum* were quantified in DNA samples from maize residues and wheat grains 2 to 3 weeks after flowering and at harvest using the qPCR method developed by Elbelt et al. (2018) [39]. Each qPCR reaction was performed in a total volume of 10 µL consisting of 1X SsoAdvanced Universal Probes Supermix, primers, and probes at concentrations described in Table 2 and 20 ng DNA (1 µL). Standard calibration curves ranged from 50 ng (C_T_ = 19.04 ± 0.82) to 0.00064 ng (C_T_ = 35.77 ± 0.83) and were performed on each PCR plate along with all samples in triplicates using a CFX96™ Real-Time System-C1000 Thermal Cycler (Bio-Rad, Hercules, CA, USA). The thermal cycle conditions were as follows: an initial denaturation step of 95 °C for 10 min followed by 40 cycles at 95 °C for 15 s then one min at the temperature specified in Table 2. Standard curve analyses, PCR efficacy, and DNA quantification were carried out using the Bio-Rad CFX Manager 3.1 software. All standard curves had an efficiency between 90 and 100% and a correlation coefficient R^2^ above 0.987. Results were expressed as ng of *Fusarium* DNA in 20 ng of total DNA.

### 2.7. Mycotoxin Extraction and Analysis

Quantitation of fifteen *Fusarium*-related mycotoxins (including DON, 15ADON, 3ADON, ZEA, NIV, MON, BEA, T-2, HT-2, ENNA, ENNA1, ENNB, ENNB1, FB1, and FB2) in wheat grains at harvest and maize residues of the four sampling stages was conducted by high-resolution liquid chromatography coupled to mass spectrometry (HR-Q-TOF LC/MS). Standards were obtained for Sigma–Aldrich, and stock solutions were prepared in DMSO to final concentrations of 0.5 mg ml^−1^. The extraction method was based on Sulyok et al. (2006) and Scarpino et al. (2019) [40,41]. Briefly, 0.5 g of dried ground grains and residues were extracted with 2 mL and 5 mL, respectively, of extraction solvent (CH_3_CN/H_2_O/CH_3_COOH, 79/20/1, *v*/*v*/*v*) by mechanical shaking at 175 rpm for 90 min (Thermoshake, Gerhardt Laboshaker, Martinsried, Germany) and subsequently centrifuged for 10 min at 5000 rpm at 10 °C on a CR3i centrifuge (Thermo Scientific, Waltham, MA, USA). The extract was diluted with the same amount of diluting solution (CH_3_CN/H_2_O/CH_3_COOH, 20/79/1, *v*/*v*/*v*) to reduce matrix effects as recommended by Scarpino et al. (2019) [41], then filtered through a 0.45 µm PTFE membrane filter (Agilent, Santa Clara, CA, USA) into an amber vial and stored at −20 °C until analysis.

Fifteen *Fusarium*-related toxin standards were combined to prepare a matrix-matched calibration curve (Appendix A) from stock solutions at 0.5 mg mL^−1^ using acetonitrile. To do so, the standard mix was diluted in a blank extract of wheat grains to obtain 12 different concentrations (1, 5, 10, 25, 50, 100, 250, 500, 1000, 2000, 2500, and 5000 ng mL^−1^). Standards were stored at −20 °C in amber vials.

Targeted mycotoxin detection and quantitation were performed using an Agilent 6530 Accurate-Mass Quadropole Time-of-Flight (Q-TOF) LC/MS system equipped with a Binary pump (1260) and degasser, a well plate autosampler set to 10 °C, and a thermostatically controlled column compartment (Santa Clara, CA, USA). Liquid chromatography separation was performed with filtered 5 µL sample aliquots at 20 °C on a Gemini^®^ C_18_ column, 100 × 2 mm, 5 µm particle size, equipped with a C_18_ 4 × 3 mm security guard cartridge (Phenomenex, Torrance, CA, USA) with a flow rate set to 0.25 mL min^−1^. The mobile phase consisted of H_2_O (eluent A) and CH_3_OH (eluent B), both of which were acidified with 0.1% *v/v* CH_3_COOH. Two separate chromatographic runs per sample were carried out to quantify all the analytes with positive (ESI+) and negative (ESI−) electrospray ionization modes, and all samples were analyzed using biological triplicates. Each run used the following gradient: mobile phase A was maintained at 90% for 4 min, then the proportion of B was increased linearly to 100% in 6 min, followed by a hold time of 5 min at 100% B before a 5 min post-run. The mass spectrometer was set with the following parameters: capillary voltage 4500 V, gas temperature 350 °C, nebulizer pressure 50 psig, drying gas 13 L min^−1^, ion range 50–1000 *m*/*z*. Injections of a quality control sample were included every twelve injections to consider any potential carry-over effects.

Toxin detection and quantitation were performed using the mean retention time ± 1 min and the corresponding quantifier and qualifier ions listed in Table 3. Toxin concentrations in samples were calculated from the equation y = mx + b, as determined by weighted (1/x2) linear regression of the matrix-matched calibration data. To validate the method, extraction recovery was evaluated in triplicate using wheat grains spiked with a standard mix at 250 and 1000 mg g^−1^. For each toxin, the limit of detection (LOD) and limit of quantification (LOQ) were assessed based on the standard deviation of the analyte response and the standard deviation slope (ICH Harmonized Tripartite Guidelines, 2005). Calibration curve calculations and mycotoxin quantitation were carried out using the Agilent MassHunter Workstation Software version B.07.01 (Santa Clara, CA, USA). Specific mycotoxin production was expressed as ng g^−1^ of lyophilized grains or residues. All compound characteristics for *Fusarium*-related mycotoxin identification and quantitation are provided in Table 3. Extracted ion chromatograms of these mycotoxins are provided in Appendix A.

### 2.8. Statistical Analyses

Statistical analyses and plotting were carried out in the R environment version 4.1.3 [42]. DNA concentrations of *Fusarium* species, as assessed by qPCR, and mycotoxin content data were log_10_ transformed to ensure homogeneity of variances. FHB disease levels 2–3 weeks after flowering and at harvest (severity index, DNA quantity, percentage of contaminated grains, and mycotoxin accumulation) were compared between fields using the Kruskal–Wallis test (*α* = 0.05) followed by a Dunn test for pairwise comparisons. Student’s *t* test was used to evaluate the difference in DNA levels of *Fusarium* species in grains and residues between flowering and harvest in two sampling years. Alpha diversity indexes (including chao1, Simpson, and equitability) were calculated with the alpha_diversity.py QIIME script using rarefied data based on the sample with the lowest number of reads. Beta-diversity analysis was calculated based on Bray–Curtis distances after Hellinger transformation of the *Fusarium* species dataset using the phyloseq package (version 1.38). Factors significantly affecting the composition and structure of *Fusarium* communities (e.g., sampling stages, years, agriculture practices, climatic conditions) were determined using multivariate homogeneity of group dispersions based on Bray–Curtis distance matrices (phyloseq) and the Adonis test in a *vegan* package (version 2.6.4). Source tracking of *Fusarium* communities in grains at flowering and harvest was performed using the SourceTracker R package (version 1.0) (https://github.com/danknights/sourcetracker, accessed on 6 September 2022) to estimate the contribution of *Fusarium* in residues and soil to grain contamination. A Sankey diagram was generated to visualize the average estimated contribution of potential sources at different stages using the networkD3 R package (version 0.4). Core taxa of three substrates (soil, residues, and grains) were assessed using the *RAM* R package (version 1.2.1.3) and defined as species or ASV present in at least 50% of samples of a given group, and a Venn diagram was created using the ggvenn package (version 0.1.10). Multi-factorial analysis (MFA) within the FactoMineR package (version 2.9) and Pearson correlations (Hmisc package version 5.1.1) were performed to assess the correlations between data groups (DNA levels, relative abundance, mycotoxin accumulation, climatic conditions). The linear regression analysis (ggscatter function in the ggpubr package version 0.6.0) was used to plot significant correlations. Significant co-occurrence patterns of *Fusarium* species per substrate were assessed using the Sparse Correlations for Compositional Data algorithm implemented in the SparCC Python module (https://github.com/bio-developer/sparcc, accessed on 12 March 2024) [43].

## 3. Results

### 3.1. FHB Development and Mycotoxin Contamination in Grains at Harvest

A very strong year effect was observed for FHB and mycotoxin contamination, as disease index, percentage of infected grains, *Fusarium* DNA quantity, and fusariotoxin accumulation in grains were significantly higher in 2021 compared with 2022 (Table 4 and Table 5). In 2021, the FHB severity index ranged from 0 to 15.45%, and 100% of grain samples at harvest were positively infected by *Fusarium* spp., predominantly by *F. graminearum*. At harvest, among the 18 wheat grain samples (6 fields × 3 samplings per field), eight were contaminated with DON, ranging from 1276 to 11,482 ng g^−1^ of dry weight. Two fields in particular (F3 and F6) were severely infected by *Fusarium* spp. in 2021, with 21% and 44% of harvested grains infected by *F. graminearum*, *F. poae*, *F. avenaceum*, and *Microdochium* spp., and up to 10,478 and 11,482 ng of DON per g of dried grains, respectively. In addition to DON, which accounted for most of the total fusariotoxin content (97.73%), grains were also contaminated by NIV (1.08%) and ENNs (1.19%, including ENNB 0.60%, ENNB1 0.12%, and ENNA 0.48%). Co-contamination of these toxins (DON, NIV, and ENNs) was observed in F6 and of two toxins (DON and ENNA) in F2, while grains in the other fields were only contaminated by DON. In 2022, FHB occurrence was lower, with the FHB severity index ranging from 0 to 1.90% at flowering. Interestingly, using the culture-dependent method, only 44.44% of grain samples at harvest were infected by *Fusarium* spp., in particular with *F. poae* (38.89%), while *F. graminearum* was not found. Levels of mycotoxins in grain samples were all below our LOD, i.e., DON (96.21 ng/g), ZEA (11.92 ng/g), 15ADON (41.69 ng/g), NIV (58.02 ng/g), ENN B (7.10 ng/g), ENN B1 (8.16 ng/g), ENN A (5.85 ng/g), ENN A1 (7.94 ng/g), FB1 (61.46 ng/g), FB1 (53.36 ng/g), MON (135.47 ng/g), BEA (9.06 ng/g), HT-2 (40.28 ng/g), and T-2 (32.52 ng/g). Using qPCR, the total DNA concentrations of three predominant *Fusarium* species (*F. graminearum*, *F. poae*, and *F. avenaceum*) were significantly lower in Y2 compared to Y1 and significantly increased in wheat grains from flowering to harvest, regardless of year (Figure 3D). In addition, the DNA quantity of *F. graminearum* in grains in 2022 was significantly lower compared to those in 2021 at both flowering (*p* < 0.05) and harvest (*p* < 0.01). In contrast, *F. poae* had significantly higher DNA levels at harvest in 2022 than in 2021 (*p* < 0.01).

### 3.2. Fusarium Diversity in Grains by EF1α Metabarcoding

During the two years of monitoring, 72 wheat head samples were analyzed to determine *Fusarium* species composition based on metabarcoding sequencing of the *EF1α* region, enabling species-level resolution of *Fusarium* spp. In total, ten *Fusarium* species were identified: *F. graminearum*, *F. poae*, *F. avenaceum*, *F. langsethiae*, *F. culmorum*, *F. equiseti*, *F. juglandicola* (recently identified—Crous et al., 2021 [44]), *F. temperatum*, *F. citricola*, and *F. flocciferum* (Figure 4A). Although *Fusarium* spp. dominance in grains varied during each growing season, *F. graminearum*, *F. poae*, and *F. avenaceum* were the most abundant species (Figure 3A–C).

In Y1 (2021), *F. graminearum* (F1, F2, and F4) and *F. poae* (F2, F3, and F6) were the predominant species at flowering, ranging from 33.43–100% and 38.09–100%, respectively, before *F. graminearum* relative abundance significantly increased at harvest (45.51–99.21%), regardless of field. In agreement with this result, harvested grains were also largely contaminated by *F. graminearum* (Table 5), in particular for F1, F3, F4, and F6, as over 10% of grain samples were infected, based on cultural data. *Fusarium avenaceum* was present in 3 out of 6 fields (F2, F4, and F5), with relative abundances ranging from 22.97 to 54.49% at harvest. The percentage of grains contaminated by *F. avenaceum* was low to moderate (0–18.00%), and no significant difference was found between fields. In Y2 (2022), *F. poae* was the predominant species, with a relative abundance ranging from 56.26–100% at both flowering and harvest. We also found a significant increase in DNA levels from flowering to harvest in Y2 (Figure 3B). One exception was found in F1, where *F. langsethiae* remained dominant at both stages (96.97% at flowering and 66.55% at harvest), whereas *F. poae* levels were low (0–15.84%). Unlike in 2021, *F. graminearum* was subdominant in 5 out of 6 fields (F1, F2, F3, F5, and F6), and low relative abundances were observed below 10%, except for F3 (34.87 ± 30.80% at flowering) (Figure 3A).

To investigate relationships between *Fusarium* spp. (relative abundance or DNA levels), climatic conditions (rain, temperature), and mycotoxin production in grains, a multiple factor analysis (MFA) was used, followed by pairwise Pearson’s correlation analysis to statistically confirm MFA-based observations. First, we found that relative abundance, determined by *EF1α* metabarcoding, and levels of *Fusarium* DNA, based on species-specific qPCR analysis, were correlated, underlying the reliability of our data. *Fusarium graminearum* was found to be negatively correlated with *F. poae* in grains in terms of relative abundance (Pearson correlation coefficient = −0.684, *p* < 0.001 and SparCC correlation coefficient = −0.287, *p* < 0.05) regardless of year, while it was positively correlated with cumulative precipitation from flowering to harvest (r = 0.713, *p* < 0.001) and mycotoxin accumulation (DON, r = 0.840; NIV, r = 0.524; ENNB, r = 0.598; ENNB1, r = 0.613; *p* < 0.05) (Figure 5). No other pairwise significant correlations involving *Fusarium* species, such as *F. avenaceum,* were obtained.

### 3.3. Fusarium Pathobiome Composition and Associated Toxins in Soils and Maize Residues

A strong substrate effect was observed both years (Figure 6) (Adonis Bray–Curtis, R^2^ = 0.358, *p* < 0.001). Interestingly, the *Fusarium* pathobiome of grains was closer to that of residues than soils. In contrast, preceding maize types (silage vs. grain) and residue management (tillage vs. minimum tillage) only had a slight influence on *Fusarium* communities (R^2^ = 0.025 and 0.026, respectively, *p* < 0.001).

In soils, twenty-three species of *Fusarium* were identified over the 2-year survey, including *F. oxysporum* as the most abundant (42.70% of mean relative abundance across sampling time), followed by *F. culmorum* (11.56%), *F. graminearum* (8.47%), *F. equiseti* (8.00%), *F. solani* (6.27%), *F. avenaceum* (5.90%), *F. flocciferum* (4.02%), *F. cerealis* (4.90%), *F. poae* (3.10%), and *F. commune* (2.19%) (Figure 4C). A significant field effect on *Fusarium* community composition was found (Adonis Bray–Curtis R^2^ = 0.188, *p* < 0.001), followed by sampling time (R^2^ = 0.120, *p* < 0.001).

In residues, twenty-four *Fusarium* species were detected by *EF1α* sequencing, including *F. graminearum* (46.40% of mean relative abundance across sampling time), *F. avenaceum* (28.324%), *F. temperatum* (10.15%), *F. cerealis* (4.80%), *F. culmorum* (3.53%), *F. poae* (2.23%), *F. equiseti* (1.08%), *F. oxysporum* (0.94%), *F. proliferatum* (0.75%), *F. sporotrichioides* (0.56%), *F. subglutinans* (0.45%), *F. flocciferum* (0.28%), *F. tricinctum* (0.12%), *F. juglandicola* (0.12%), *F. solani* (0.04%), *F. praegraminearum* (0.03%), *F. meridionale* (0.03%), *F. venenatum* (0.03%), *F. acuminatum* (0.03%), *F. commune* (0.02%), *F. bulbicola* (0.01%), *F. lacertarum* (0.01%), *F. verticillioides* (0.01%), *F. anthophilum* (0.003%). The two predominant *Fusarium* spp. in grains (*F. graminearum* and *F. avenaceum*) were also predominant in residues at both stages and for both years (Figure 4B). In contrast, *F. poae*, although dominant in grains (especially in 2021), was only detected in a few residue samples (46 out of 124) and at very low relative abundance (mean 2.23%). In agreement, *F. poae* DNA levels were also significantly lower in residues compared to those of *F. graminearum* and *F. avenaceum,* with similar levels throughout the wheat cycle (Figure 7). A significantly moderate effect on *Fusarium* community composition in residues was found between fields (Adonis Bray-Curtis R^2^ = 0.206, *p* < 0.001) and sampling time (R^2^ = 0.125, *p* < 0.001).

In addition to *Fusarium* composition, mycotoxin accumulation was also quantified in residues across time (Figure 8), resulting in 83 out of 132 residue samples contaminated by multiple mycotoxins above LOQs: DON, 15ADON, ZEA, FB1, FB2, ENNB, and NIV. Significantly strong effects of sampling stages (Adonis Bray-Curtis R^2^ = 0.232, *p* < 0.001) and two-way interactions between fields and sampling stages (R^2^ = 0.458, *p* < 0.001) were observed (Figure 9). The sampling year had a very low effect on the mycotoxin profile of residues (R^2^ = 0.046, *p* < 0.05), while preceding maize types and residue management had no effect (*p* > 0.05). Fusariotoxin diversity was higher in residues compared to grains, including DON (representing 35.17% of quantified toxins, only detected after maize harvest and wheat seedling), ZEA (28.63%), 15ADON (11.65%, only detected after maize harvest), FB1 (14.07%), FB2 (9.17%), ENNB (0.87%, only in Y2 after maize harvest and at seedling), and NIV (0.44%, only in F6Y1 at seedling). Toxin accumulation was usually found to be highest just after maize harvest, then tended to decrease over time, in particular for DON and 15ADON (Figure 8). High concentrations of DON and 15ADON were found at T1 (after the harvest of maize) and significantly decreased after several months. Among targeted toxins, ZEA, FB1, and FB2 were the only ones that were systematically present at different stages, with variation between fields and years. We observed a significant increase of ZEA at T2 (seedling) or T3 (2–3 weeks after wheat flowering) in 3 out of 12 fields (F3Y1, F5Y1, and F1Y2). Levels of fumonisins significantly increased from T1 to T3 and/or T4 (wheat harvest) in 8 out of 12 fields (F2, F4, F5 in Y1 and F1, F2, F3, F5, and F6 in Y2), while there was a significant increase from T1 to T2 for the other fields. We also found a positive correlation between DON and 15ADON (r = 0.809, *p* < 0.001), or ZEA (0.472, *p* < 0.001), and between FB1 and FB2 (r = 0.916, *p* < 0.001). Interestingly, F6Y1 at seedling (T2) was the only residue sample contaminated by NIV and showed the highest relative abundance of *F. poae* (88.31%). Note that mycotoxin-free residues mean residues contaminated with mycotoxins at levels inferior to LOD.

*Fusarium* pathobiomes from residues, soil, and grains were compared to determine the core species of the three substrates, found in at least 50% of samples of each group (Figure 4D). Four species (*F. avenaceum*, *F. temperatum*, *F. equiseti*, and *F. cerealis*) and *F. solani* were uniquely found in residues and soil, respectively. *F. poae* was identified as the unique species of wheat grains, while *F. oxysporum* and *F. culmorum* were shared between residue and soil substrates, and *F. graminearum* was the only core species of all three substrates.

### 3.4. Fusarium Dispersal from Residues to Grains

Transfer of *Fusarium* inoculum from maize residues and soil to wheat grains was assessed by determining the contribution of inoculum sources (residues or soils) in contaminating wheat heads based on Source Tracker and *EF1α* sequencing. In Y1, 83.80 ± 15.28% of *Fusarium* ASVs in grains were similar to those from residues, while soils only contributed to 19.60% of ASVs in grains (and only at harvest). In contrast, contributions of ASVs from residues to grains were much lower in Y2 (ANOVA F = 18.07, *p* < 0.001) (Figure 10), at both flowering and harvest, while soils and an unknown source potentially contributed to grain contaminations at higher levels in 2022 compared to 2021 (*p* < 0.05).

Overall, residues were found to be the primary inoculum source of *Fusarium* in wheat heads, but other substrates, including soil and unknown sources, also contributed to grain contamination, especially under low disease pressure. Besides, although *Fusarium* spp. loads in residues remained similar across time and fields, grains in 2021 were more severely infected than in 2022, suggesting that other factors contributed to the FHB risk, including climatic conditions or agronomic practices.

### 3.5. Climatic and Agronomic Factors Influencing Grain Contamination with Fusarium

Climatic conditions, including cumulative daily average temperature (°C) and cumulative precipitation from one week before flowering to harvest (mm), were monitored in 2021 and 2022. Significantly higher levels of precipitation (157.02 mm ± 13.27 in 2021 versus 80.87 mm ± 19.70 in 2022) and lower temperatures (881.52 °C days ± 105.78 versus 958.33 °C days ± 37.57) were recorded in 2021 compared to the drier and hotter weather in 2022 (Figure 11). Cumulative precipitations were positively correlated to *F. graminearum* abundance in grains (r = 0.713, *p* < 0.001) and DON contamination (r = 0.645, *p* < 0.01) and negatively correlated to *F. poae* contamination (r = −0.694, *p* < 0.001). In addition to climatic conditions during flowering, other factors should be considered, in particular the number and timing of fungicide applications during flowering as well as cultivar varieties, which could potentially create confounding effects with weather conditions on grain contamination. Overall, the number of fungicide applications (between 2 and 3) and levels of resistance to FHB (ranging from 3.5 to 5.5 [34]) were similar across fields (Table 1).

## 4. Discussion

This study provides a comprehensive description of the diversity and dynamics of *Fusarium* communities and their associated mycotoxins in three key agronomic substrates known to be colonized by *Fusarium* spp.: wheat grains, crop residues, and soil. We focused our study on six wheat fields (Brittany, France) monitored for two consecutive years with four sampling dates per year during the wheat cycle. FHB levels, mycotoxin contents, and individual *Fusarium* spp. abundances in grains, determined by *EF1α* metabarcoding or qPCR, varied according to substrates, fields, and years. Under high disease pressure conditions encountered in 2021, wheat grains were dominated by *F. graminearum,* while in 2022, when disease pressure was low, *F. poae* outcompeted *F. graminearum*. Total precipitation from one week before wheat flowering to harvest was identified as the primary factor promoting *F. graminearum*, DON, and ENN contamination in grains. Our results align with previous studies showing a high positive correlation between levels of *F. graminearum,* trichothecenes, and precipitation during flowering, the most susceptible stage for FHB infection in wheat [45,46] and barley [47]. The timing of the wheat anthesis is indeed crucial for predicting the FHB risk. Late-planting increases the risk of infection because climatic conditions during wheat anthesis are expected to coincide with more favorable conditions for disease development [48,49]. In 2022, there was no effect of the sowing date on FHB incidence and severity, likely due to the hot and dry weather from May to July covering the flowering period across fields. In contrast, in 2021, we found a significant reduction of *Fusarium* infection and DON contamination in F4 (early sowing field) compared to F6 (late sowing field, three weeks later), both fields located in the same city. The amount of precipitation during flowering was lower in F4 (8.8 mm) compared to F6 (43.2 mm), aligning with the known climatic preference of *F. graminearum*.

Interestingly, a higher occurrence of *F. poae* compared to that of *F. graminearum* was observed in spring 2022, characterized by low levels of precipitation during wheat flowering, resulting in low disease pressure and low levels of fusariotoxins in grains at harvest. The optimal growth conditions of *F. graminearum* and *F. poae* were reported to be different under field and greenhouse conditions. *F. graminearum* was associated with warm and humid conditions, whereas *F. poae* was associated with relatively drier and hotter weather [8,50,51,52]. Our observations are consistent with reports showing increased *F. poae* prevalence during dry and warm years, when conditions are unfavorable for *F. graminearum* development [53,54,55]. Specifically, in 2022, *F. poae* significantly increased during grain filling and maturity, while *F. graminearum* remained at low levels. Conversely, in 2021, although *F. poae* and *F. graminearum* levels were similar at flowering, only *F. graminearum* grew until harvest. Such observation only partly corroborates with the hypothesis proposed by Audenaert et al. (2009), who state that *F. poae* acts as a secondary invader, colonizing the wheat heads already weakened by more aggressive FHB pathogens [56]. In our study, we found that *F. poae* could thrive independently in grains when conditions are unfavorable for more aggressive pathogens such as *F. graminearum*. This interaction between *F. poae* and *F. graminearum* had significant implications for mycotoxin contamination. While DON was the most frequently detected mycotoxin, NIV was only detected in the two most severely infected fields in 2021. However, we cannot rule out the fact that samples could be contaminated by NIV at levels below our LOD (58.02 ng/g). The NIV-contaminated grains contained significantly higher DNA quantities of both *F. graminearum* and *F. poae*, which are potential NIV producers [57]. Given that the *F. graminearum* NIV chemotype is relatively rare in France [58], it seems more likely that *F. poae* was responsible for the NIV contamination in these fields. Furthermore, a high relative abundance of *F. langsethiae,* a morphologically closely related species of *F. poae* [59], was found in two out of six fields in 2022, while a minor occurrence was noted in 2021. Such as *F. poae*, *F. langsethiae*, a weaker pathogen compared to *F. graminearum*, is known as a type A trichothecene producer (HT-2 and T-2 toxins) in oats, wheat, or barley [60,61,62,63]. However, trichothecenes type A were not detected in wheat grains at levels above 40.29 and 32.52 ng/g, respectively, regardless of contamination by *F. langsethiae* (note that it was recently demonstrated that *F. poae* should be unrelated to T2 or HT2 production, contrary to what is often stated in the literature [64]). In addition to *F. graminearum* and *F. poae*, *F. avenaceum* was detected as a predominant species in certain fields. Frequent co-occurrences of *F. graminearum*, *F. avenaceum*, and *F. poae* have been widely reported on small-grain cereals in France [63,65]. Despite being FHB causal agents, *F. poae* and *F. avenaceum* were reported to cause milder FHB symptoms compared to *F. graminearum*, aligning with our observations [65]. Additionally, ENN contamination was documented in various small-grain cereals in France, with *F. tricinctum* as the main contributor, followed by *F. avenaceum* and *F. poae* to a much lesser extent [63]. In our survey, co-contamination of *F. graminearum* with *F. avenaceum* and/or *F. poae* in grains was the rule rather than the exception. Yet, this resulted in grains being co-contaminated with DON, ENN, and/or NIV only in two fields (F2Y1 and F6Y1). Even if mycotoxin co-occurrence is identified in a limited number of fields, this contamination pattern poses a serious concern for the food chain.

Apart from precipitation during wheat flowering, the absence of tillage and the presence of previous host crop residues are known to increase the FHB risk. Fields under tillage are less likely to be contaminated with mycotoxins [22]. *Fusarium* spp. can survive saprophytically on cereal crops over the winter before contaminating novel hosts in the following season [22,27]. Our results in 2021 suggested that *Fusarium* communities in residues significantly contributed to wheat grain contamination, in particular by *F. graminearum*, at both flowering and harvest when weather conditions were conducive to disease development. These conditions may be linked to rain-splash dispersal. In contrast, the contribution of residues to grain contamination was much lower in 2022, when the climate was hotter and drier, despite similar *Fusarium* inoculum loads in residues both years. Conidia produced by *F. graminearum*, *F. avenaceum*, and *F. poae* have also been shown to be vertically transported from residues to wheat heads by rain drops or overhead irrigation [27,66,67]. Reduced rainfall likely decreased pathogen inoculum migration from crop residues, thus reducing overall FHB infection and mycotoxin levels in wheat grains. Interestingly, *F. poae* was predominantly detected in grains both years and was also found at low levels in residues and in soil samples, suggesting these two substrates were probably not major sources of *F. poae* inoculum. Previous studies have also indicated that both crop residues and gramineous weeds were probably not an important source of *F. poae* [18,68,69]. Likewise, the *F. langsethiae* infection observed in wheat grains in 2022 might have originated from neighboring fields due to its absence in residues and low abundance in soil. Other potential sources of inoculum, including air, wild grasses, or the plant itself, need to be investigated further, although this species has never been reported as a wheat plant endophyte or in seed-borne diseases to date. Unlike residues, soil was not an important source of primary inoculum, and *F. graminearum*, *F. poae*, and *F. avenaceum* were not considered predominant species in soil. The dominance of *F. oxysporum* in soil is consistent with its nature as a soil-borne fungus [70,71]. Although several studies demonstrated the correlation between the density and distribution of infected residues in the field and FHB contamination, no clear effect of preceding maize types (maize silage vs. grain) was observed in this survey. Other factors, such as the resistance levels of wheat cultivars to FHB and fungicide application at flowering (including both timing and number of applications) [72], also probably contributed to the infection success of wheat heads with *Fusarium* species. Yet, our study was not designed to specifically investigate the impact of these factors, all in the same range of magnitude, on *Fusarium* contamination.

Another aspect of our study was to thoroughly describe the diversity of *Fusarium* and mycotoxin contamination over the course of maize residue degradation on the soil surface. The presence of mycotoxins in the soil and drainage water and their interactions with the soil fauna have been reported in only a few studies [33,73,74]. This raises concerns related to the impact of these toxic metabolites in soils and water and how they might contaminate the food chain or impact the local ecosystem. *Fusarium*-infected plants or residues can be considered the main contributors to mycotoxin accumulation in soils since these compounds can be washed off the residues entering the soil [33]. In addition, given the high levels of multiple mycotoxin co-occurrences in maize residues, the portions of maize plants that are harvested and used as silage for cattle and pig feeds are likely contaminated with these mycotoxins, posing a feed safety threat. Mycotoxins, in particular DON, ZEA, fumonisins, ochratoxins, and enniatin B, are usually detected in maize silage and maize plants before ensiling [75,76]. DON has been frequently found in corn silages worldwide, including in the Netherlands (maximum concentration of 3142 µg/kg), Poland (7860 µg/kg), Germany (3944 µg/kg), and the United States (5100 µg/kg). Similarly, fumonisins represent the most frequent mycotoxin in feed, with more than a 50% prevalence rate and up to 2490 µg/kg in corn silage [76]. Given the high levels of mycotoxins (despite our moderately high LOQ) and their diversity in our study, the risk associated with such environmental and feed contaminations requires further evaluation.

Similar to the studies of Köhl et al. (2007), Landschoot et al. (2011), Cobo-Díaz et al. (2019), and Mourelos et al. (2024), *F. graminearum* was found to be predominant in our maize residue samples, with a mean relative abundance of 46.40% [18,69,77,78]. This species was able to survive in wheat fields for a long period, from maize harvest time to wheat harvest, during the ongoing degradation of residues throughout the sampling period. In addition, twenty-four other *Fusarium* species were detected in maize residues during the 2-year field monitoring. Among these, only *F. graminearum*, *F. poae*, *F. avenaceum*, *F. culmorum*, *F. equiseti*, *F. juglandicola*, *F. temperatum*, and *F. flocciferum* were also found in wheat grains. This suggests that some species are more specific to maize substrate, such as fumonisin producers (*F. verticillioides*, *F. proliferatum*, and *F. oxysporum*), which are known maize-associated pathogens. In line with our study, several previous reports indicated that maize ears and stalks were colonized by a substantially greater number of *Fusarium* species compared to wheat grains and other small grain cereals [5,79].

The coexistence of various *Fusarium* species on maize residues, including stalks, leaves, and ears, led to multi-fusariotoxin contaminations such as trichothecenes (both type A and B), zearalenone, and fumonisins [4,80]. This co-contamination suggests potential synergistic toxicity and raises questions about why these fungi produce toxic metabolites and their role in a given ecosystem. Although saprophytic growth constitutes a major part of the life cycle of *Fusarium* species and contributes to inoculum formation [81,82], knowledge of this phase and the biological role of mycotoxin production is still limited. This is mainly due to the fact that most research on fungal pathogens focuses on living plant-pathogen interactions. Mycotoxins, in particular trichothecenes and fumonisins, are known to enhance pathogen virulence, facilitating the colonization of plant tissues before the subsequent saprophytic stage [82,83]. In our residue samples, we found co-occurrences of DON and 15ADON, as well as FB1 and FB2, consistent with frequent co-contamination detected in maize silage [84] and whole-plant harvested maize [85]. Variations in mycotoxin accumulation between different fields and years suggest influences of microclimate, microbial communities, and other organisms present in the ecosystems [86]. The decrease in DON and 15ADON quantities in maize residues over time aligns with the low DON production during the *Fusarium* saprophytic phase compared to their pathogenic phase [83]. The variation of trichothecene quantities in residues may also be linked to the saprophytic survival strategies of different *Fusarium* species, as in vitro and in planta studies have shown the impact of competitive interactions among *Fusarium* species on their production of mycotoxins [82,87,88,89]. Mycotoxins, including DON and ZEA, are also hypothesized to contribute to niche competition on crop residues by weakening competitors such as other fungi, bacteria, nematodes, protozoa, and soil fauna during saprophytic growth [90]. Conversely, soil fauna contributes to suppressing pathogen inoculum and degrading their mycotoxins [91]. *Fusarium* communities seem to be an attractive food source for collembolans and earthworms because of their high nutritional value compared to other soil fungi [92]. The significant degradation of DON over time was also reported during the treatment of anecic earthworms on wheat straw [74,93], indicating the potential of soil fauna for sustainable mycotoxin control in fields to mitigate the risk of environmental pollution. In addition, the disappearance of trichothecenes might be attributed to residue decomposition [94] and mycotoxin degradation or mineralization by soil microbiota [94,95,96,97,98]. The production of mycotoxins by *Fusarium* species on residues may also be linked to their life cycle dynamics, particularly in response to environmental cues and stressors. For instance, several studies underscored the regulatory role of G protein signaling pathways in coordinating both mycotoxin production and the formation of spores [99,100]. Although there were fluctuations in the levels of zearalenone and fumonisins across different stages of residue degradation, our results highlighted an increase in these metabolites at seedling and during the grain filling period. These periods coincide with critical phases in the *Fusarium* life cycle and environmental conditions, e.g., the winter seedling stage and wheat flowering (FHB infection period). The mycotoxins produced during these stages likely serve multiple functions, including pathogen survival under stress conditions and enhancing the inoculum for infection during favorable environmental periods [100]. This hypothesis is also supported by the relationship between mycotoxin production and the sporulation of *Fusarium* species [99,100]. Throughout the degradation of crop residues, mycotoxin production by *Fusarium* communities in crop residues varied due to the complex influence of a multitude of both biotic and abiotic factors. Given the high levels of mycotoxins detected in maize residues during the wheat cycle and their detrimental effects as environmental pollutants, further investigations are required to elucidate their specific biological roles during the saprophytic growth of *Fusarium* on crop residues as well as their contribution during the infectious stage.

## 5. Conclusions

Based on our results, we demonstrated substantial *Fusarium* species diversity and significant shifts among communities in wheat grains during the grain-fill and maturity periods over a two-year monitoring period. Given the known variability among *Fusarium* species in terms of pathogenicity and mycotoxin production profiles, our metabarcoding approach provided a much more exhaustive description of the *Fusarium* community, which is sometimes lacking in current studies. This approach can complement traditional techniques based on isolation or species-specific PCR identifications/quantifications. The analysis of crop residues and soil highlighted the significant role of infected residues in contaminating wheat heads with *Fusarium* species. This contamination potentially occurs through rain splashing at the field scale or long-distance spore transfer by wind. However, crop residues were not the only source of infection, as evidenced by the low abundance of *F. poae* in residues despite its dominance in wheat heads. The next step will be to determine whether the strains of *Fusarium* species in crop residues, in particular *F. graminearum*, are drawn from the same populations as those causing FHB in wheat grains at flowering and harvest. Population-level analyses will be required to understand the relationships between predominant *Fusarium* species on different hosts and their migration patterns. Our study also revealed a high diversity of *Fusarium* species and the co-contamination of multiple mycotoxins in crop residues across the wheat cycle, corroborating previous reports. These findings raise questions about the ecological roles of mycotoxins during fungal residue colonization, including their potential impacts on microbiota and gain in competitiveness for the producing fungus. Further research should explore these aspects to improve our understanding of *Fusarium* ecology and its impact on agricultural systems.

## Figures and Tables

**Figure 1 microorganisms-12-01783-f001:**
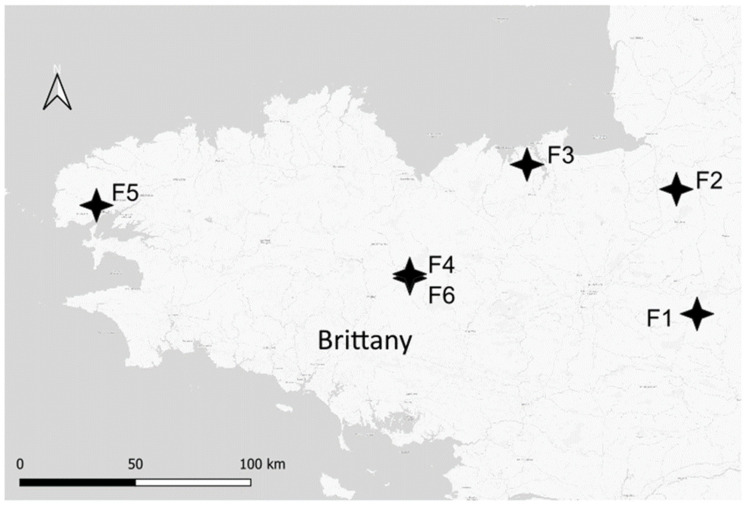
Map depicting the locations of wheat fields monitored in 2020–2021 and 2021–2022 in Brittany, France.

**Figure 2 microorganisms-12-01783-f002:**
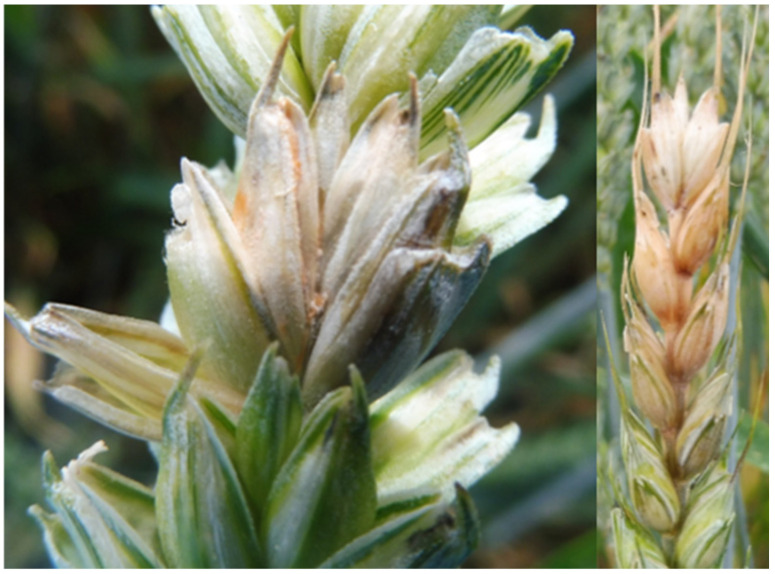
Typical Fusarium head blight symptoms on wheat heads 2–3 weeks after flowering.

**Figure 3 microorganisms-12-01783-f003:**
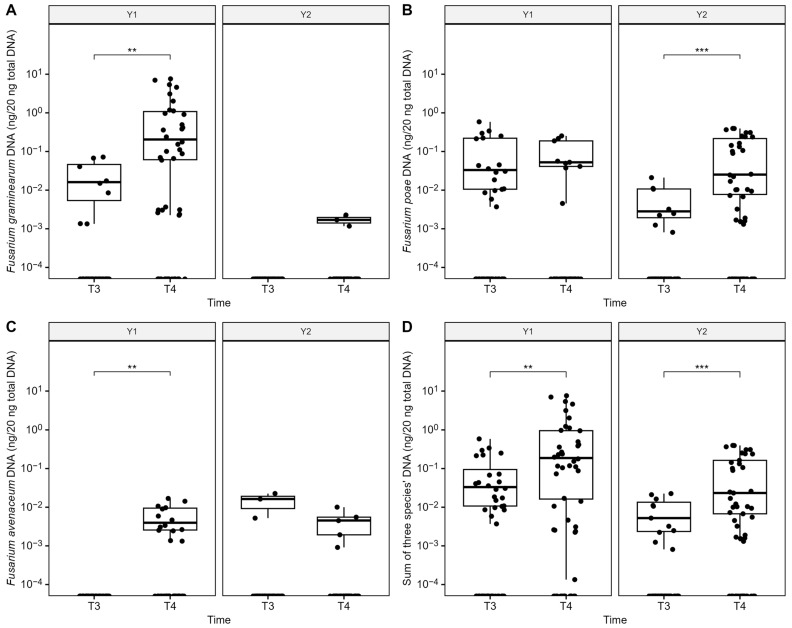
Levels of DNA (ng/20 ng total DNA) in log scale of the three dominant *Fusarium* species (*F. graminearum*: (**A**), *F. poae*: (**B**), *F. avenaceum*: (**C**), and total: (**D**)) in grains at flowering (T3) and harvest (T4) during a two-year monitoring survey (Y1 and Y2) in 6 fields. Asterisks represent significance levels based on a *t*-test, ** = 0.01, and *** = 0.001.

**Figure 4 microorganisms-12-01783-f004:**
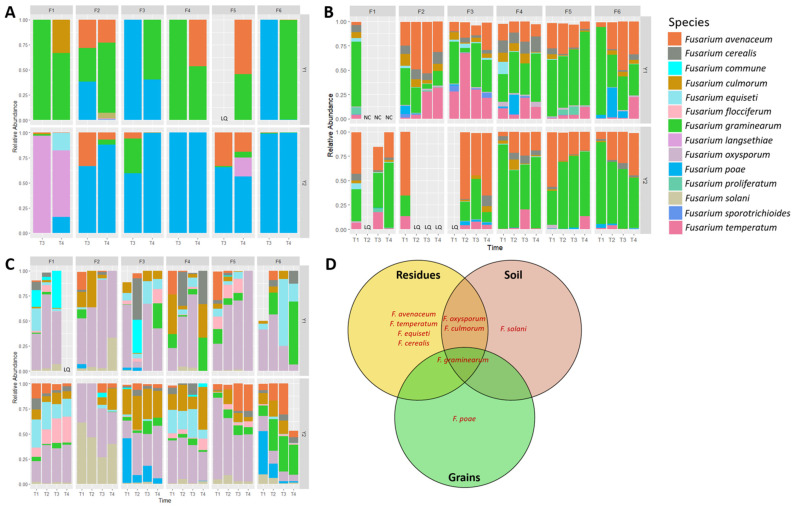
Histograms showing the top 10 *Fusarium* spp. in grains (**A**), residues (**B**), and soil (**C**) across wheat cycle during a 2-year monitoring survey (Y1 and Y2) in 6 fields (F1 to F6) at four sampling stages (T1: after harvest of maize, T2: seedling, T3: 2–3 weeks after flowering, and T4: harvest). NC (Not Collected): no residues were visible, thus collected, after tillage in F1 in 2021. LQ (Low Quality): quality of sequences obtained from these samples were too low and excluded from analysis. Relative abundance was calculated based on the mean of three sampling points per field. Venn diagram (**D**) showing the unique and shared *Fusarium* species among residues, soil, and grains, with the minimum percentage of samples in each substrate set at 50%.

**Figure 5 microorganisms-12-01783-f005:**
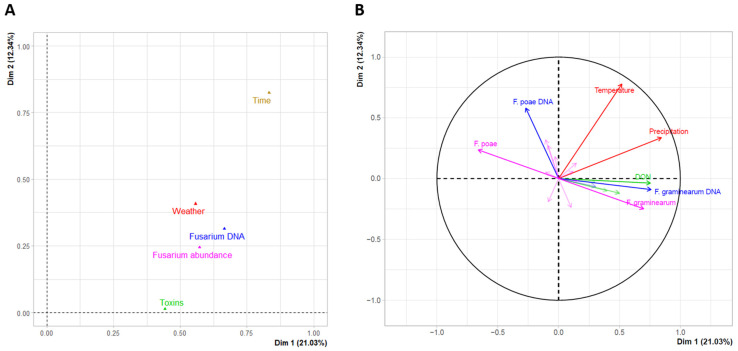
Multiple factor analysis (MFA) of variations in the *Fusarium* communities and mycotoxin production in wheat grains from flowering to harvest over a two-year monitoring period. (**A**) Individual factor map showing the interrelationships of variables. (**B**) Correlation circle plot showing the correlations between *Fusarium* DNA levels (blue), *Fusarium* relative abundance (pink), fusariotoxin accumulation in grains (green), and weather conditions (red). Only variables with a contribution (cos^2^) above 0.4 were labeled.

**Figure 6 microorganisms-12-01783-f006:**
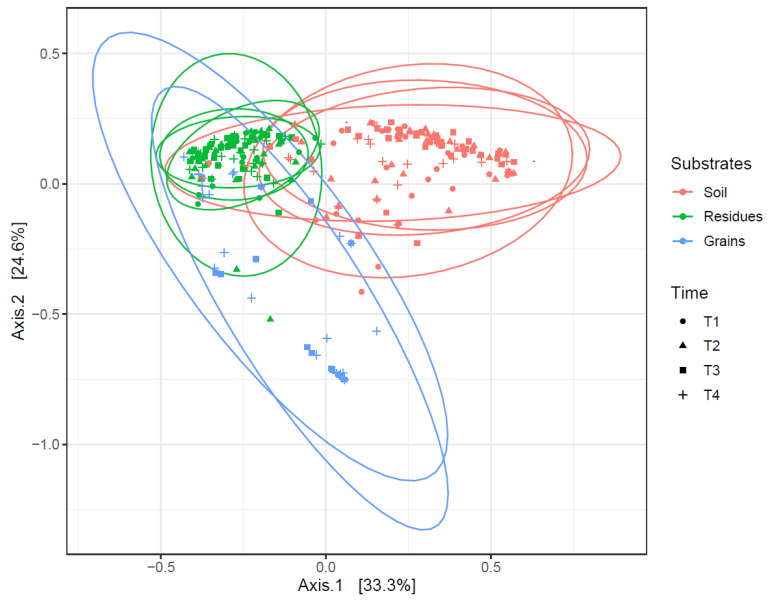
PCoA of *Fusarium* communities in soil (red), residue (green), and grain (blue) samples at different stages (T1 to T4) based on Bray–Curtis distances. Significant difference between *Fusarium* communities on three substrates: Adonis Bray–Curtis R^2^ = 0.3580 (*p* value < 0.001). T1: after harvest of maize; T2: seedling; T3: 2–3 weeks after flowering; T4: harvest.

**Figure 7 microorganisms-12-01783-f007:**
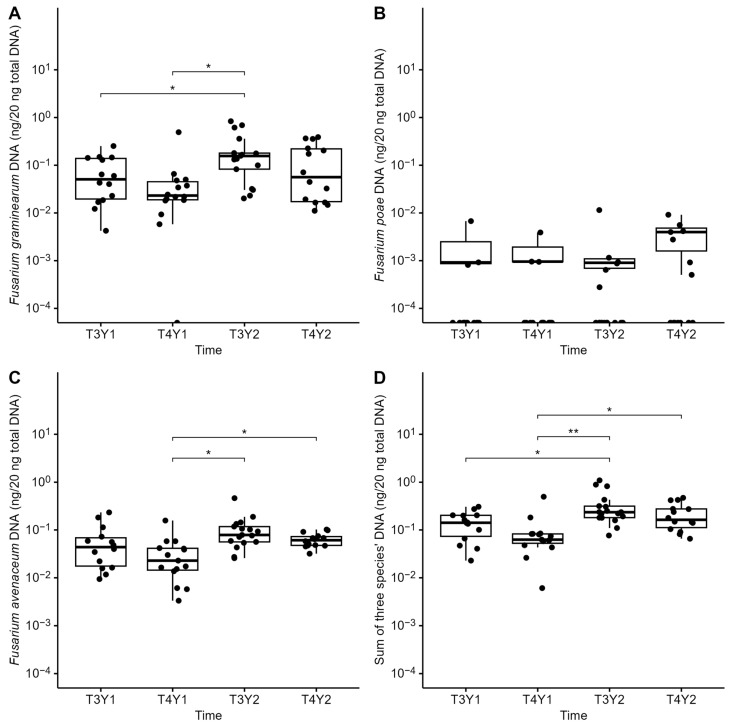
DNA concentration (ng/20 ng DNA total) of three dominant *Fusarium* species (*F. graminearum*: (**A**), *F. poae*: (**B**), *F. avenaceum*: (**C**), and sum of DNA of the three species: (**D**)) in residues at flowering (T3) and harvest (T4) during two years of monitoring (Y1 and Y2). Asterisks (* and **) represent significance levels at 5 and 1 difference by *t*-test respectively.

**Figure 8 microorganisms-12-01783-f008:**
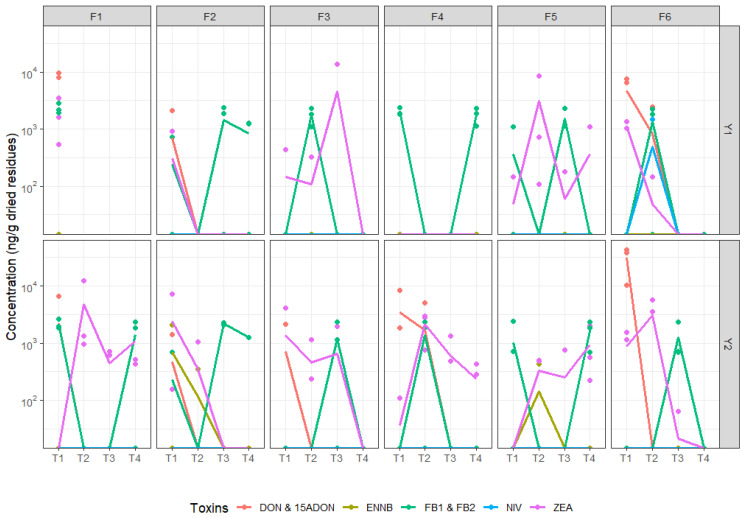
Mycotoxin dynamics in log scale in residues collected from six wheat fields at four sampling stages (post-harvest of maize T1, seedling T2, 2–3 weeks after flowering T3, and harvest T4) of two-year monitoring (2021—Y1 and 2022—Y2). No residues were collected in F1 after the harvest of maize because the field was tilled. Dots represent levels of toxins (ng/g dried grains) for each of the three repetitions, and lines represent the mean values. Limits of detection and limits of quantification of each toxin were as follows: zearalenone (11.92 and 36.13 ng/g), deoxynivalenol (96.21 and 291.53 ng/g), 15ADON (41.69 and 126.33 ng/g), nivalenol (58.02 and 175.81 ng/g), enniatin B (7.10 and 21.51 ng/g), enniatin B1 (8.16 and 24.74 ng/g), enniatin A (5.85 and 17.73 ng/g), enniatin A1 (7.94 and 24.07 ng/g), fumonisin B1 (61.46 and 186.25 ng/g), fumonisin B2 (53.36 and 161.70 ng/g), moniliformin (135.47 and 410.50 ng/g), beauvericin (9.06 and 27.46 ng/g), HT-2 (40.28 and 122.06 ng/g), and T-2 (32.52 and 98.55 ng/g).

**Figure 9 microorganisms-12-01783-f009:**
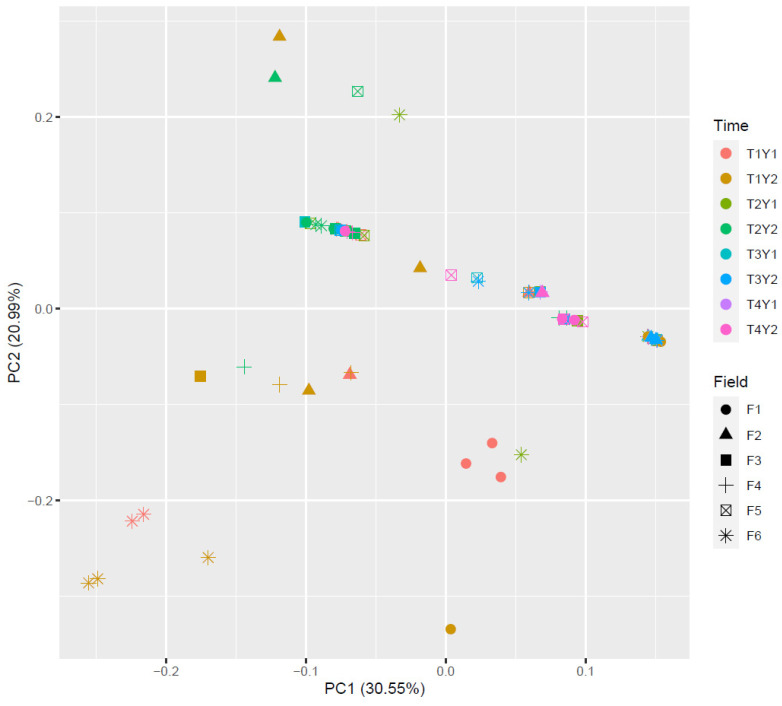
Principal component analysis (PCA) of fusariotoxins quantified in residues according to fields (F1 to F6), sampling stages (T1 to T4), and year of monitoring (Y1 and Y2). T1: after harvest of maize; T2: seedling; T3: 2–3 weeks after flowering; T4: harvest.

**Figure 10 microorganisms-12-01783-f010:**
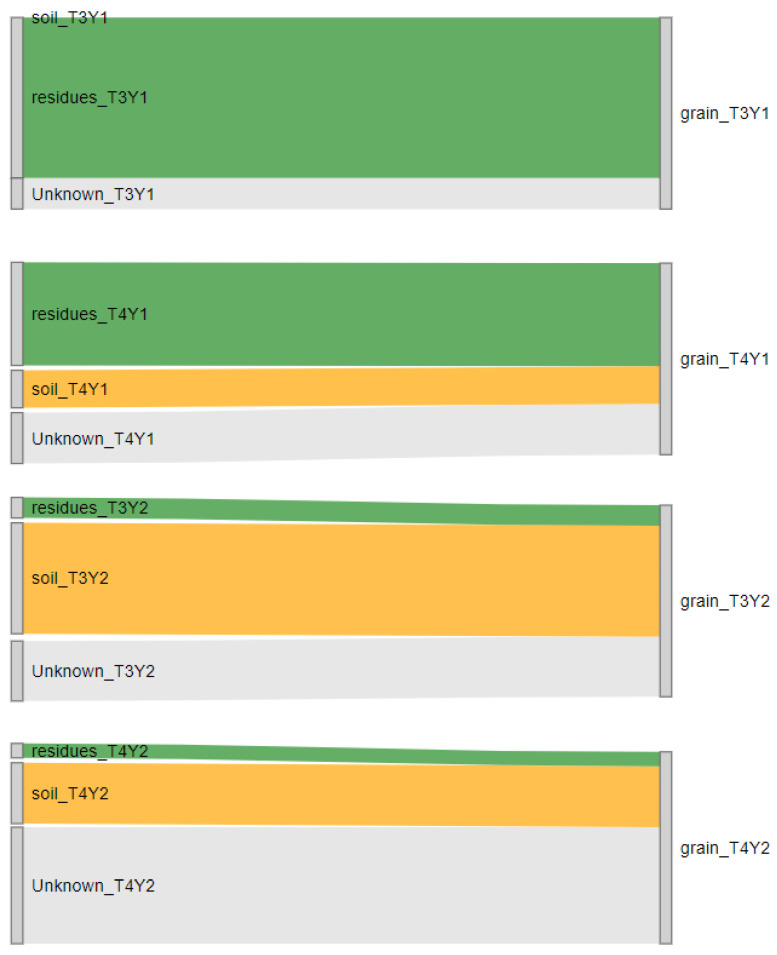
Sankey diagram of source tracking of *Fusarium* spp. in grains at flowering (T3) and harvest (T4) during 2-year monitoring (Y1 and Y2). Mean contributions of residues (green), soil (yellow), and an unknown source (gray) were shown.

**Figure 11 microorganisms-12-01783-f011:**
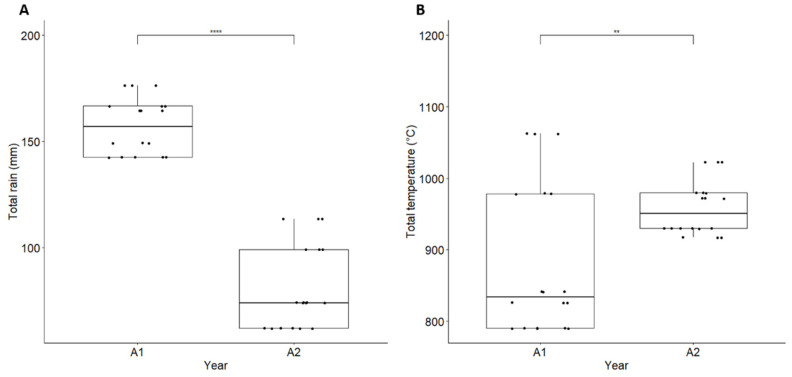
Cumulative daily precipitation (mm) (**A**) and temperature (°C) (**B**)at six wheat fields from flowering to harvest of two-year monitoring. Asterisks (** and ****) represent the significance level at 1 and 0.01% difference by *t*-test, respectively.

**Table 1 microorganisms-12-01783-t001:** Details of sampling fields of two-year monitoring study of Fusarium Head Blight.

Field	Location	Wheat Cycle	Previous Crop: Maize Silage or Grain	Maize Cultivar	Maize Residue Management	Wheat Cultivar	Wheat Sowing Date	Resistance Level to DON *	Wheat Flowering Date	Number of Fungicide Applications before Complete Flowering
F1Y1	Gennes sur Seiche, Ille-Et-Vilaine	2020–2021	Maize silage	LBS 2964 et DKC 3450	Tillage	Complice	5 November 2020	3.5	20 May 2021	3
F2Y1	Villamée, Ille-Et-Vilaine	2020–2021	Maize grain	Belami	Min-till	Vyckor, LG Absalon	17 October 2020	Vyckor 5.5 LG Absalon 5	3 June 2021	3
F3Y1	Pleslin-Trigavou, Côtes d’Amor	2020–2021	Maize grain	Kolossalis	Min-till	KWS Extase	3 November 2020	4	31 May 2021	1
F4Y1	Loudéac, Côtes d’Amor	2020–2021	Maize silage	Pontivy	Min-till	Vyckor	29 October 2020	5.5	1 June 2021	3
F5Y1	Guilers, Finistère	2020–2021	Maize silage	Franceen, Kilomeris, LG 31.295	Min-till	Chevignon, KWS Extasse,	7 November 2020	Chevignon 5 KWS Extasse 4 RGT Perkussio 3.5	21 May 2021	3
						RGT Perkussio				
F6Y1	Loudéac, Côtes d’Amor	2020–2021	Maize grain	RGT Synfonixx	Min-till	Fructidor, Vyckor	20 November 2020	Fructidor 5 Vyckor 5.5	4 June 2021	3
F1Y2	Gennes sur Seiche, Ille-Et-Vilaine	2021–2022	Maize silage	Franceen	Min-till	Complice	29 October 2021	3.5	15 May 2022	2
F2Y2	Villamée, Ille-Et-Vilaine	2021–2022	Maize grain	Kolossalis kws	Tillage	LG Absalon	16 November 2021	5	21 May 2022	1
F3Y2	Plelin-Trigavou, Côtes d’Amor	2021–2022	Maize grain	DKC3888	Min-till	Hyking	28 October 2021	4	18 May 2022	1
F4Y2	Loudéac, Côtes d’Amor	2021–2022	Maize silage	LG 31.272	Min-till	Fructidor	27 November 2021	5	25 May 2022	0
F5Y2	Milizac, Finistère	2021–2022	Maize silage	Figaro, Milkmax, LG 31.259	Min-till	Chevignon, Winner, Vyckor	10 November 2021	Chevignon 5 Winner 4.5 Vyckor 5.5	20 May 2022	2
F6Y2	Loudéac, Côtes d’Amor	2021–2022	Maize grain	RGT Synfonixx	Min-till	Vyckor, Chevignon, Fructidor	18 November 2021	Vyckor 5.5 Chevignon 5 Fructidor 5	25 May 2022	2

* Levels of resistance to risk of deoxynivalenol (DON) according to ARVALIS (2021) [34].

**Table 2 microorganisms-12-01783-t002:** Primer and probe sequences and qPCR conditions for *F. graminearum*, *F. poae*, and *F. avenaceum* (Elbelt et al., 2018) [39].

	Nom	Sequence	Reporter/ Quencher (5′/3′)	Final Concentration (nM)	Annealing Temperature (°C)
*F. graminearum*	EF1-FCFG_F	TCGATACGCGCCTGTTACC	FAM/TAMRA	300	62
	EF1-FG_R	ATGAGCGCCCAGGGAATG		300	
	grami2-EF1_rev	AGCCCCACCGGGAAAAAAATTACGACA		100	
*F. poae*	EF1-FP2_F	CTCGAGCGATTGCATTTCTTT	FAM/TAMRA	300	60
	EF1_FP2_R	GGCTTCCTATTGACAGGTGGTT		300	
	EF1-FP	CGCGAATCGTCACGTGTCAATCAGTT		100	
*F. avenaceum*	EF1-FA_F2	CATCTTGCTAACTCTTGACAGACCG	FAM/TAMRA	300	64
	EF1-FA_R3	GGGTAATGAATGCGTTTCGAA		300	
	EF1-FA	AGCGAGTCGTGGGAATCGATGGG		150	

**Table 3 microorganisms-12-01783-t003:** *Fusarium*-related mycotoxin quantitation parameters used for HR-QTOF-LC/MS analysis. LOD: limit of detection; LOQ: limit of quantification; R^2^: regression coefficient.

Mycotoxin	Retention Time (Min)	Quantifier		Qualifier		Extraction Recovery (%)	Range (ng/g)	R^2^	LOD (ng/g)	LOQ (ng/g)
		Ion	*m*/*z*	Ion	*m*/*z*					
ZEA	14.06	[M-H]^−^	317.1394	/	/	91.18 ± 6.10	1–250	0.978	11.92	36.13
DON	4.65	[M+CH_3_COO]^−^	355.1398	[M-H]^−^	295.1187	126.01 ± 20.57	100–2000	0.989	96.21	291.53
15ADON/3ADON	12.07	[M-H]^−^	337.1293	[M+CH_3_COO]^−^	397.1504	91.26 ± 10.57	100–2000	0.998	41.69	126.33
NIV	2.31	[M+CH_3_COO]^−^	371.1348	[M-H]^−^	311.1136	131.77 ± 2.84	50–1000	0.985	58.02	175.81
ENN B	14.70	[M+NH_4_]^+^	657.4433	[M+Na]^+^	662.3987	111.96 ± 10.48	1–500	0.998	7.10	21.51
ENN B1	14.82	[M+NH_4_]^+^	671.4590	[M+Na]^+^	676.4144	118.16 ± 13.47	1–500	0.997	8.16	24.74
ENN A	15.05	[M+NH_4_]^+^	699.4903	[M+Na]^+^	704.4457	105.03 ± 10.02	1–500	0.998	5.85	17.73
ENN A1	14.94	[M+NH_4_]^+^	685.4746	[M+Na]^+^	690.4300	108.85 ± 14.29	1–500	0.997	7.94	24.07
FB1	13.33	[M+H]+	722.3957	[M+Na]^+^	744.3777	80.73 ± 1.26	1–2500	0.993	61.46	186.25
FB2	13.74	[M+H]+	706.4008	[M+Na]^+^	728.3828	95.89 ± 4.89	1–2500	0.995	53.36	161.70
MON	2.63	[M-H]^−^	96.9931	/	/	119.87 ± 2.68	100–2000	0.980	135.47	410.50
BEA	14.79	[M+NH_4_]^+^	801.4483	[M+Na]^+^	806.3987	154.07 ± 15.92	1–250	0.987	9.06	27.46
HT-2	13.47	[M+Na]^+^	447.1989	/	/	96.63 ± 3.50	1–2000	0.992	40.28	122.06
T-2	13.71	[M+Na]^+^	489.2095	[M+NH_4_]^+^	484.2541	69.45 ± 18.83	5–500	0.965	32.52	98.55

ZEA: zearalenone; DON: Deoxynivalenol; 15ADON: 15-acetyl-deoxynivalenol; 3ADON: 3-acetyl-deoxynivalenol; NIV: nivalenol; ENNB: enniatin B; ENNB1: enniatin B1; ENNA: enniatin A; ENNA1: enniatin A1; FB1: fumonisin B1; FB2: fumonisin B2; MON: moniliformin; BEA: beauvericin; HT-2: HT-2 toxin; T-2: T-2 toxin.

**Table 4 microorganisms-12-01783-t004:** Fusarium Head Blight symptoms and *Fusarium* spp. DNA levels 2 to 3 weeks after flowering. Different letters indicate significant differences between groups (*p* < 0.05). *Fg*: *Fusarium graminearum*; *Fp*: *F. poae*; *Fa*: *F. avenaceum*.

Year	Field	Severity Index (%)	Levels of *Fusarium* DNA(Log10 (1 + DNA) pg/20 ng DNA)		
			*Fg*	*Fp*	*Fa*
Y1	F1Y1	0.00 ± 0.00 ^a^	0.000 ± 0.000 ^a^	0.000 ± 0.000 ^a^	0.000 ± 0.000 ^a^
	F2Y1	2.05 ± 1.23 ^bd^	0.000 ± 0.000 ^a^	0.000 ± 0.000 ^a^	0.000 ± 0.000 ^a^
	F3Y1	10.90 ± 3.96 ^d^	0.000 ± 0.000 ^a^	1.102 ± 0.277 ^bc^	0.000 ± 0.000 ^a^
	F4Y1	1.35 ± 0.61 ^bcd^	0.973 ± 0.785 ^b^	0.000 ± 0.000 ^a^	0.000 ± 0.000 ^a^
	F5Y1	0.44 ± 0.43 ^ac^	0.000 ± 0.000 ^a^	0.000 ± 0.000 ^a^	0.000 ± 0.000 ^a^
	F6Y1	2.30 ± 1.25 ^bd^	0.082 ± 0.164 ^ab^	2.192 ± 0.444 ^c^	0.000 ± 0.000 ^a^
Y2	F1Y2	0.82 ± 0.91 ^abc^	0.000 ± 0.000 ^a^	0.000 ± 0.000 ^a^	0.000 ± 0.000 ^a^
	F2Y2	0.75 ± 0.48 ^abc^	0.000 ± 0.000 ^a^	0.000 ± 0.000 ^a^	0.000 ± 0.000 ^a^
	F3Y2	0.45 ± 0.15 ^ac^	0.000 ± 0.000 ^a^	0.387 ± 0.585 ^ab^	0.000 ± 0.000 ^a^
	F4Y2	1.03 ± 0.59 ^abc^	0.000 ± 0.000 ^a^	0.158 ± 0.256 ^ab^	0.000 ± 0.000 ^a^
	F5Y2	1.37 ± 0.58 ^bcd^	0.000 ± 0.000 ^a^	0.000 ± 0.000 ^a^	0.378 ± 0.587 a
	F6Y2	1.33 ± 0.51 ^bcd^	0.000 ± 0.000 ^a^	0.096 ± 0.194 ^ab^	0.000 ± 0.000 ^a^

**Table 5 microorganisms-12-01783-t005:** Fusarium Head Blight symptoms, *Fusarium* DNA levels, and toxin accumulation at harvest. Different letters indicate significant differences between groups (*p* < 0.05). *Fg*: *Fusarium graminearum*; *Fp*: *F. poae*; *Fa*: *F. avenaceum*; *Micro*: *Microdochium* spp.; DON: deoxynivalenol; NIV: nivalenol; ENNB: enniatin B; ENNB1: enniatin B1; ENNA1: enniatin A1; ENNA: enniatin A; ZEA: zearalenone.

Year	Field	Percentage of Grains Contaminated by				*Levels of Fusarium DNA* (Log10 (1 + DNA) pg/20 ng DNA)			Toxin Accumulation (ng g^−1^ Dry Weight)						
		*Fg*	*Fp*	*Fa*	*Micro*	*Fg*	*Fp*	*Fa*	DON	NIV	ENNB	ENNB1	ENNA1	ENNA	ZEA
Y1	F1Y1	18.67 ± 4.16 ^ab^	0.00 ± 0.00 ^a^	4.00 ± 4.00 ^a^	7.33 ± 2.31 ^a^	0.724 ± 1.092 ^ab^	0.000 ± 0.000 ^a^	0.145 ± 0.291 ^a^	425.35 ± 638.03 ^ab^	<LOD	<LOD	<LOD	<LOD	<LOD	<LOQ
	F2Y1	4.00 ± 3.46 ^ac^	0.00 ± 0.00 ^a^	0.67 ± 1.15 ^a^	0.00 ± 0.00 ^b^	0.696 ± 0.956 ^abc^	0.000 ± 0.000 ^a^	0.000 ± 0.000 ^a^	847.81 ± 1271.71 ^ab^	<LOD	<LOD	<LOD	<LOD	80.84 ± 7.68	<LOQ
	F3Y1	14.67 ± 8.33 ^ab^	4.67 ± 1.15 ^b^	1.33 ± 2.31 ^a^	0.00 ± 0.00 ^b^	2.080 ± 1.361 ^bc^	1.250 ± 1.061 ^abc^	0.318 ± 0.480 ^a^	4493.61 ± 4672.66 ^bc^	<LOD	<LOQ	<LOD	<LOD	<LOD	<LOQ
	F4Y1	19.33 ± 1.15 ^ab^	1.33 ± 1.15 ^ab^	8.67 ± 9.02 ^a^	10.00 ± 4.00 ^a^	0.742 ± 1.128 ^ab^	0.000 ± 0.000 ^a^	0.559 ± 0.513 ^a^	1570.11 ± 2355.17 ^ab^	<LOD	<LOQ	<LOQ	<LOQ	<LOD	<LOQ
	F5Y1	2.67 ± 2.31 ^ac^	0.00 ± 0.00 ^a^	0.67 ± 1.15 ^a^	2.00 ± 2.00 ^ab^	0.062 ± 0.186 ^a^	0.000 ± 0.000 ^a^	0.000 ± 0.000 ^a^	<LOD	<LOD	<LOD	<LOD	<LOD	<LOD	<LOD
	F6Y1	39.33 ± 11.02 ^b^	1.33 ± 1.15 ^ab^	2.00 ± 3.46 ^a^	2.00 ± 0.00 ^a^	3.117 ± 0.392 ^c^	0.546 ± 0.819 ^ac^	0.172 ± 0.267 ^a^	9247.49 ± 1857.51 ^c^	183.05 ± 274.58	101.37 ± 116.48	20.22 ± 15.47	<LOQ	<LOD	<LOQ
Y2	F1Y2	0.00 ± 0.00 ^c^	0.00 ± 0.00 ^a^	0.00 ± 0.00 ^a^	0.00 ± 0.00 ^b^	0.000 ± 0.000 ^a^	0.000 ± 0.000 ^a^	0.000 ± 0.000 ^a^	<LOD	<LOD	<LOD	<LOQ	<LOQ	<LOD	<LOQ
	F2Y2	0.00 ± 0.00 ^c^	0.51 ± 0.89 ^a^	0.00 ± 0.00 ^a^	0.00 ± 0.00 ^b^	0.000 ± 0.000 ^a^	0.555 ± 0.558 ^abc^	0.000 ± 0.000 ^a^	<LOD	<LOD	<LOD	<LOD	<LOD	<LOD	<LOD
	F3Y2	0.00 ± 0.00 ^c^	2.56 ± 2.35 ^ab^	0.51 ± 0.89 ^a^	0.00 ± 0.00 ^b^	0.105 ± 0.210 ^a^	2.066 ± 0.810 ^b^	0.000 ± 0.000 ^a^	<LOD	<LOD	<LOD	<LOD	<LOD	<LOD	<LOD
	F4Y2	0.00 ± 0.00 ^c^	0.00 ± 0.00 ^a^	0.00 ± 0.00 ^a^	0.00 ± 0.00 ^b^	0.000 ± 0.000 ^a^	0.115 ± 0.235 ^a^	0.289 ± 0.440 ^a^	<LOD	<LOD	<LOD	<LOD	<LOQ	<LOD	<LOQ
	F5Y2	0.00 ± 0.00 ^c^	1.54 ± 2.67 ^ab^	0.00 ± 0.00 ^a^	0.00 ± 0.00 ^b^	0.000 ± 0.000 ^a^	0.490 ± 0.504 ^abc^	0.052 ± 0.156 ^a^	<LOD	<LOD	<LOD	<LOD	<LOD	<LOD	<LOD
	F6Y2	0.00 ± 0.00 ^c^	4.62 ± 2.67 ^b^	0.00 ± 0.00 ^a^	0.00 ± 0.00 ^b^	0.037 ± 0.112 ^a^	1.945 ± 0.577 ^bc^	0.031 ± 0.094 ^a^	<LOD	<LOD	<LOD	<LOD	<LOD	<LOD	<LOD

Limits of detection and limits of quantification of each toxin were as follows: zearalenone (11.92 and 36.13 ng/g), deoxynivalenol (96.21 and 291.53 ng/g), 15ADON (41.69 and 126.33 ng/g), nivalenol (58.02 and 175.81 ng/g), enniatin B (7.10 and 21.51 ng/g), enniatin B1 (8.16 and 24.74 ng/g), enniatin A (5.85 and 17.73 ng/g), enniatin A1 (7.94 and 24.07 ng/g), fumonisin B1 (61.46 and 186.25 ng/g), fumonisin B2 (53.36 and 161.70 ng/g), moniliformin (135.47 and 410.50 ng/g), beauvericin (9.06 and 27.46 ng/g), HT-2 (40.28 and 122.06 ng/g), and T-2 (32.52 and 98.55 ng/g).

## Data Availability

The raw reads of the *EF1α* metabarcoding presented in this study have been deposited at the NCBI and are openly available under the Bioproject PRJNA1132884.

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
