# Peer review of "Shifts in Fusarium Communities and Mycotoxins in Maize Residues, Soils, and Wheat Grains throughout the Wheat Cycle: Implications for Fusarium Head Blight Epidemiology"

_microorganisms, 2024, doi:10.3390/microorganisms12091783_

Round 1
Reviewer 1 Report
Comments and Suggestions for Authors
General comments:
I have reviewed the manuscript entitled “Shifts in Fusarium communities and mycotoxins in maize residues, soils, and wheat grains during the wheat cycle: implications for Fusarium head blight epidemiology (microorganisms-3126218)” carefully. In this work, the authors monitored six minimum tillage wheat fields in Brittany, France over two years. The results indicated substantial Fusarium species diversity and significant shifts among the communities in wheat grains during the grain-fill and maturity period.
Overall, the purpose of the study is clear and the writing of the manuscript is fine. However, there are obvious drawbacks in mycotoxin detection.
I only have minor comment to this manuscript.
Specific comments:
1. The uniform should be followed in the whole text, or example the citations in the main text, such as Cobo Diaz et al. (2019) on Line 181, Figaro [37] on Line 185, Elbelt et al. (2018), (Elbelt et al. 2018) (page 8), Sulyok et al. (2006) and Scarpino et al. (2019) [40,41], etc.
2. The authors have very bad LOD and LOQ for most of the Fusarium toxins quantitative analysis. Here I strongly request the authors to improve the detection method and provide the original data about calibration curves made with known amounts of standard compounds, the recovery rates of each toxin in each substrate, as well as the matrix effects data as supplementary materials, which are essential for quantification assays. All these information is also essential for readers to repeat it. Besides, all the results and discussion related to mycotoxin contamination must be re-prepared, including Figures and Tables showed in the main text. In my opinion, the current version is not acceptable to me.
3. The last paragraph on page 19, I request the authors to double check exactly how many Fusarium species detected carefully. I can only find 24 species in this section.
Reviewer 2 Report
Comments and Suggestions for Authors
The manuscript evaluate Fusarium Head Blight epidemiology relating soil and wheat grains during the wheat cycle about Fusarium species and related mycotoxins. They adopted advanced methods to identify and quantify micota and mycotoxins and also good statistic tools to show simultaneous effect of different variable. Some relations were previously showed in several scientific report. The innovation are the analytical technic and the statistic treatment that were well used by the authors to reinforce knowledge about FHB.
Line 228-229: it is need to review the phrase: "The extract was diluted with the same amount of diluting solution (CH3CN/H2O/CH3COOH, 20/79/1, v/v/v) to reduce matrix effects then filtered through 0.45 µm PTFE membrane filter xxxxx" To dilute the solution is not the best way to reduce matrix effects. What about a matrix curve for solve the effects?
Percentage units have different numbers of decimal digits. It is necessary to standardize. The same comments for other unities.
Line 448-449 "These findings raise questions about the biological roles of mycotoxins during the saprophytic stage growth in residues as well as their effects on soil microbiota." This comment is not so clear, I suggest include this point in deep in the discussion item.
Round 2
Reviewer 1 Report
Comments and Suggestions for Authors
To some extent, I agree with the authors‘ response, but I still recommend that the authors provide chromatograms of 15 Fusarium toxins in standard solutions as supplementary materials.
Author Response
Comment 1: To some extent, I agree with the authors‘ response, but I still recommend that the authors provide chromatograms of 15 Fusarium toxins in standard solutions as supplementary materials.
Response 1: Thank you for your time and understanding. As requested, chromatograms of the 15 mycotoxins are now included as supplementary materials (Fig. S2 cited in the text L. 266-267).